# Quality of Life after Bariatric Surgery—A Systematic Review

**DOI:** 10.3390/ijerph19159078

**Published:** 2022-07-26

**Authors:** Regina Sierżantowicz, Jerzy Robert Ładny, Jolanta Lewko

**Affiliations:** 1Department of Surgical Nursing, Medical University of Bialystok, 15-274 Bialystok, Poland; 2Department of General and Endocrine Surgery, Medical University of Bialystok, 15-276 Bialystok, Poland; ladnyjr@wp.pl; 3Department of Primary Health Care, Medical University of Bialystok, 15-054 Bialystok, Poland; jola.lewko@wp.pl

**Keywords:** bariatric surgery, obesity, quality of life, SF-36, BAROS

## Abstract

Background: Most studies analyzing the health-related quality of life (HRQOL) after bariatric treatment ceased at five years post-surgery or even earlier, and it is unclear whether the HRQOL benefit persists for a longer time. This paper reviews sparse evidence regarding HRQOL in patients who underwent bariatric surgery at least nine years prior. Materials and Methods: A of PubMed, Scopus and Google Scholar between 2007–2021 was carried out for the studies investigating HRQOL as an outcome measure in patients after bariatric surgery of any type and having at least a 9-year follow-up. Inconsistent reporting of weight loss or postgraduate study results unrelated to QoL were not included in the study. The study used the PICO procedure. Results: The review of 18 identified publications demonstrated that bariatric treatment seems to provide a persistent benefit in terms of HRQOL, especially its physical component score. Due to psychological predispositions, some patients appear to be less likely to benefit from bariatric treatment, whether in terms of HRQOL or bodyweight reduction. Inconsistent and imprecise studies may limit the evidence included in a review. Conclusions: The early identification of such patients and providing them with holistic care, including psychological intervention, would likely further improve the outcomes of bariatric treatment.

## 1. Introduction

Obesity, defined as body mass index (BMI) ≥ 30 kg/m^2^, without a doubt constitutes a serious public health issue. The number of adults with obesity is estimated at 650 million or 13% worldwide; this figure is three times higher than in 1975 [1]. Obesity is a part of the so-called metabolic syndrome (MS) and frequently coexists with other components of MS, such as type 2 diabetes mellitus and arterial hypertension. Additionally, obesity carries the risk of cardiovascular diseases, some malignancies and premature death. Excess body weight was also demonstrated to be associated with depression and severely deteriorated quality of life [2,3,4,5,6,7].

While public health campaigns, such as dietary intervention and lifestyle modifications, play an unquestioned role in obesity prevention, they are not necessarily effective in persons with already developed obesity, especially severe and complex (BMI ≥ 40 kg/m^2^ or BMI 35–40 kg/m^2^ with another significant health problem linked to excess body weight) [8]. In the cases mentioned above, bariatric surgery remains a vital treatment option, as it was demonstrated not only to result in greater body weight loss but also to provide better control of diabetes than lifestyle interventions or pharmacotherapy alone [9,10]. The most commonly performed types of bariatric surgeries are sleeve gastrectomy and Roux-en-Y gastric banding (RYGB) [11].

Aside from somatic consequences, obesity is also known to exert a detrimental effect on the patient’s wellbeing. Persons with obesity were shown to frequently suffer from social stigma, depression, anxiety and eating disorders and often presented with a distorted body image [12,13,14,15,16,17,18,19,20]. All these conditions contribute to the deterioration of health-related quality of life (HRQOL), commonly defined as “a multidomain concept that represents the patient’s general perception of the impact of an illness and its treatment on physical, psychological, and social aspects of life” [21]. Therefore, HRQOL and other patient-reported outcomes are increasingly recognized as important endpoints in research on the effectiveness of bariatric treatment [22]. It should be remembered that objective clinical outcome measures, such as excess weight loss (EWL), do not necessarily accurately reflect the subjective feelings of the patient [23]. Moreover, a post-treatment improvement in the patient’s HRQOL should not be considered merely as an additional effectiveness measure but also recognized as an important determinant of further compliance with the physician’s recommendations [24].

Generally, the impact of bariatric treatment can be measured in the context of generic and obesity-specific HRQOL [22]. The most commonly used measure of generic HRQOL in bariatric patients is the SF-36 questionnaire [25,26,27,28,29,30,31,32,33]. This instrument consists of 36 items forming eight domains (physical functioning, role-physical, bodily pain, general health, vitality, social functioning, role emotional and mental health); the domains can be analyzed separately or cumulatively as a physical component score (PCS) and mental component score (MCS) [34,35]. Unlike individual domains, PCS and MCS were demonstrated to have adequate validity in the population of patients with severe obesity, and as such, were recommended as primary endpoints in the studies of generic HRQOL after bariatric treatment [36].

A limitation of the SF-36 stems from the fact that it does not measure some obesity-specific aspects, such as body image and social stigma [37]. To fill in this gap, an instrument measuring obesity-specific HRQOL is also recommended to be used in the research on bariatric treatment outcomes. One of the most popular obesity-specific HRQOL scales is the Impact of Weight on Quality of Life—Lite (IWQOL-Lite) instrument [20]. This scale consists of 31 items grouped into five domains (physical function, self-esteem, sexual life, public distress, and work) [38]. The scale was developed and validated for use in people with severe obesity [38,39]. Recently, a shorter version with 20 items, IWQOL-Lite Clinical Trials (CT), was developed, compliant with the US Food and Drug Administration guidance for patient-reported outcomes [40]. Another instrument used commonly, albeit primarily in studies with a shorter follow-up period, is the Bariatric Analysis and Reporting Outcome System (BAROS), developed in 1998 [41]. The BAROS is a validated scoring system based on a combination of physician-reported outcomes (EWL), changes in medical conditions and patient-reported outcomes in terms of the quality of life. The latter are scored on a Likert scale ranging from −0.50 to +0.50. The higher the overall score, obtained as a combination of EWL, improvement in medical conditions, data on postoperative complications and reoperations, if any, and HRQOL ratings, the better the outcome. A score of 1 or less is considered a “failure”, whereas >7 points is considered an “excellent” outcome [41].

Published reviews suggest that HRQOL, whether generic or obesity-specific, improves within 1–2 years after bariatric treatment and then again deteriorates, but is still better than before treatment [26,28,30]. Unfortunately, most of the source studies analyzing the problem in question ceased at year 5 post-surgery or even earlier, primarily because of problems in following up with operated patients in a longer perspective [22]. The research question is whether the HRQOL benefit persists over a longer period of time. An answer to this question would not only constitute an additional argument for the use of bariatric treatment in patients with severe and complex obesity but might also have some important implications in the context of long-term holistic management of such patients.

The aim of this paper is to review sparse evidence regarding HRQOL in patients who underwent bariatric surgery at least nine years prior.

## 2. Materials and Methods

The study was carried out in accordance with the recommendations in the PRISMA guidelines [42]. A complete checklist, according to the PRISMA statement, is reported in Appendix A. A systematic search of PubMed, Scopus and Google Scholar was carried out. All published studies investigating HRQOL as an outcome measure in patients after bariatric surgery of any type were identified. The search limits were defined as ‘English’ (language), and ‘the beginning of a given database through 31 October 2021’ (publication date) between 2007–2021. The study used the PICO procedure, taking into account the obesity with BMI > 30 kg/m^2^ and qualification for the bariatric procedure, performing a bariatric procedure, treatment in the control group of a non-bariatric procedure and the result, i.e., assessment of quality of life and weight loss. All randomized clinical trials and prospective, retrospective, cross-sectional and observational studies conducted in human patients receiving bariatric treatment (1), with HRQOL (2) and related parameters (body image, suicidality, social relationships, etc.) considered as outcome measures at least nine years after the procedure (3) and following patients with obesity (4) were included in the review. Inconsistent reporting of weight loss or postgraduate study results but not related to QoL was not included in the study. The keywords used to search for the data were bariatric surgery, obesity, and quality of life, and patients were observed for ≥9 years.

Review papers, case reports and studies we were unable to extract for full-text review were excluded.

During the first stage, all records were identified from searches of the electronic databases and duplicates were removed (RS). During the second stage, two researchers (JRŁ and JL) independently screened the titles and abstracts of 2304 identified articles to find potentially eligible studies. During the third stage, potentially eligible studies were selected for full-text review. Finally, additional publications were identified based on a manual search of the reference lists of the previously selected papers. Disagreements were resolved by mutual consent after discussion (Figure 1).

### Quality Assessment

Methodological quality assessment of the included studies was performed by 2 authors (RS and JRŁ) independently according to the Newcastle–Ottawa Scale (NOS) on different types of studies (randomized and non-randomized clinical trial, prospective cohort study, observational study, mail and phone survey). Included studies were assessed on items, including representativeness, patient selection, ascertainment, mention of conflict of interest, comparability, outcome assessment, follow-up length, and follow-up adequacy, with a maximum score of 9. Comparative studies with a score ≥ 6 were considered of high quality (low risk of bias), while studies with a score < 6 were considered of moderate or low quality (high risk of bias).

## 3. Results

A total of eighteen published studies were identified, with ten among them analyzing HRQOL as an outcome measure (with additional surrogate measures or without) and four solely with the outcome measures considered surrogate endpoints of HRQOL (Table 1).

### 3.1. Studies Analyzing HRQOL

Assessment HRQOL after surgical and conventional intervention for severe obesity was performed by Karlsson et al. [43]. This Swedish Obese Subjects (SOS) controlled longitudinal trial was completed by a total of 1276 patients, with among them 655 treated surgically and 621 who received conventional treatment. The study patients were followed up with for ten years, with HRQOL determined at 0.5, 1, 2, 3, 4, 6, 8 and 10 years. HRQOL was measured with a battery of generic and obesity-specific measures, including the Current Health (CH) scale, the short version of the Mood Adjective Check List (MACL), Hospital Anxiety and Depression Scale (HADS), Social Interaction category from the Sickness Impact Profile (SIP) and Obesity-Related Problems Scale (OP). In the surgical group, peak HRQOL improvements were observed during the first year post-procedure, followed by a gradual deterioration between years 1 and 6 and relative stabilization between years 6 and 10. However, at ten years, net gains were observed in all HRQOL domains compared with the baseline. Meanwhile, no clear pattern could be identified in the HRQOL of the conventional treatment group, with some minor improvements observed within the initial two years post-intervention, some parameters improving no earlier than the end of the observation period, and some not improving at all or even deteriorating [43].

Kinzl et al. [44] examined the long-term consequences of laparoscopic gastric banding in morbidly obese patients. After a mean follow-up of 10 years (range 9–12 years), a set of questionnaires was sent to 180 previously operated-on patients. HRQOL was determined with two questionnaires: BAROS and Bariatric Quality of Life Index (BQL). The response rate was 62%. Based on the BAROS scores, 14% of the respondents graded their outcomes “excellent”, 31% “very good”, 23% “good”, 14% “fair” and 18% “failure”. Analysis of variance demonstrated that postoperative BAROS scores correlated significantly with both EWL and the extent of weight loss. HRQOL measured with BQL was significantly higher than before the treatment. A post-treatment improvement in HRQOL was found in 78% of the respondents, whereas 11% reported no change and another 11% reported deterioration of post-procedure HRQOL. The improvement in BQL scores correlated closely with the satisfaction with weight loss [44]. O’Brien et al. [45] conducted a randomized clinical trial to compare the effects of laparoscopic adjustable gastric banding (LAGB) and intensive medical weight loss in treating mild to moderate obesity. A total of 80 patients with a BMI of 30–35 kg/m^2^ were randomized to the surgical and non-surgical program and followed up for ten years. The HRQOL was measured with the SF-36. At ten years of follow-up, 37 patients were available for the analysis, among them 27 from the surgical group and 10 from the non-surgical group. At two years post-procedure, the surgical group presented with significantly higher MCS values than at the baseline, and this beneficial effect also persisted at ten years. However, no significant changes were observed in PCS values for the surgical patients. Nevertheless, both MCS and PCS values at ten years did not differ significantly from respective Australian norms. Additionally, at two years post-procedure, PCS values in the surgical patients were significantly higher than in the non-surgical group [45]. Aarts et al. [46] used BAROS to analyze the long-term outcomes of LAGB in a group of 201 patients who received this procedure for morbid obesity between 1995 and 2003. The group was followed up for a mean of 13.6 years. Out of 193 patients evaluable at 14 years, 88 had still their band in place. In approximately half of patients from this subset (51%), BAROS score corresponded to “failure”, whereas “fair”, “good”, “very good” and “excellent” outcomes were reported by 19%, 22%, 7% and 1% of the respondents, respectively [46]. Canetti et al. [47] compared HRQOL of 36 bariatric surgery (Silastic ring vertical-banded gastroplasty) patients and 34 participants of a weight-loss program at one and ten years after the intervention. HRQOL was assessed with the SF-36. Additionally, the respondents completed the Mental Health Inventory (MHI), Neuroticism Scale of the NEO Five-Factor Inventory, Fear of Intimacy Scale and the Shapiro Control Inventory. Both PCS and MCS values improved significantly within the first year after the surgery and then deteriorated significantly between years 1 and 10. At the end of the follow-up period, PCS values were still significantly higher than before the surgery, whereas the MCS values reverted to the baseline level. Additionally, at ten years after the intervention, surgical patients showed a significant deterioration of mental health status measured with MHI scores, with an insignificant improvement over the first year, followed by a significant deterioration at year 10. Additionally, the other three psychological parameters, neuroticism, fear of intimacy and sense of control, followed a similar pattern, with an initial improvement followed by a significant deterioration below the baseline levels. In the dieting group, PCS and MCS values at year 10 did not differ significantly from their baseline levels, and no significant changes were observed over time in the values of MHI, neuroticism, fear of intimacy and sense of control levels [47]. Herpertz et al. [48] conducted a prospective longitudinal cohort study to analyze the HRQOL and psychological functioning of German patients within nine years after restrictive surgical treatment for obesity. The study included 152 patients undergoing restrictive surgical treatment, 249 persons participating in conventional weight reduction program and 128 obese controls without any weight-loss treatment. HRQOL was determined with the SF-36 at 1, 2, 4 and 9 years. Additionally, the patients were assessed for anxiety and depressive symptoms with HADS and for global self-esteem with the Rosenberg Self-Esteem Scale (RSES). PCS values in the surgical group improved up to four years post-surgery and then decreased by year 9 but were still significantly better than at the baseline. Regardless of the analyzed time point, the surgical group reported greater improvements in PCS compared with the other groups. Meanwhile, MCS values in the surgical group improved significantly by year 4 and then returned to the baseline level at year 9. After controlling for the baseline scores, the surgical group experienced greater impairment of MSC than the other two groups. Additionally, anxiety, depression and self-esteem scores for the surgical group followed a similar pattern, with an initial improvement followed by a return to the baseline levels at year 9 [48]. Aasprang et al. [49] analyzed long-term HRQOL in patients with severe obesity who underwent biliopancreatic diversion with duodenal switch (BPD-DS) at a single Norwegian center. The prospective cohort study included 50 patients, followed up for ten years. HRQOL was measured with the SF-36 at the baseline and 1, 2, 5 and 10 years post-procedure. A total of 35 patients were available at ten years. Both PCS and MCS values of the patients increased significantly from baseline to year 10. A mixed-effect model analysis demonstrated that the effect sizes for PCS and MCS compared to the norm population adjusted for BMI, age and gender were large and moderate, respectively. A significant improvement in most of the SF-36 domain scores was documented as well. However, the scores at ten years were still below the normative values for the Norwegian population. While a change in BMI between the baseline and five years did not correlate significantly with either PCS or MCS change, significant correlations were found between an increase in BMI at 5–10 years and reductions in PCS and MCS [49]. Another study analyzed HRQOL in 43 patients who had undergone biliopancreatic diversion with duodenal switch (BPD/DS) in 1999–2010. The patients were followed up for nine years post-procedure. The analyzed HRQOL measures included SF-36 and IWQOL-Lite; additionally, the patients completed the Beck Depression Inventory (BDI). A total of 30 patients were available at year 9. The SF-36 survey showed that prior to the surgery, the study patients presented with marked impairment of all eight domains. One year after the procedure, the scores normalized at the level corresponding to community norms, and this improvement was also maintained at year 9. The changes in IWQOL-Lite scores followed a similar pattern; the scores for all domains, severely impaired preoperatively, normalized at one year post-procedure and remained within the normal levels until the end of the follow-up period. The only exception was the reported assessment of sexual life, which did not reach the community norms. BDI scores were available for a subset of 38 patients. Prior to BPD/DS, the BDI scores of the respondents corresponded to moderate depression; the scores improved significantly within one year post-procedure and remained unchanged for up to nine years [50]. Nguyen et al. [51] conducted a randomized clinical trial to compare the outcomes of laparoscopic gastric bypass and laparoscopic gastric banding in 197 patients with BMI between 35 and 60 kg/m^2^. One of the outcome measures was HRQOL determined with the SF-36. The study patients were followed up for ten years. Regardless of the procedure type, the SF-36 scores improved significantly from the baseline in all eight domains; additionally, significant increases in PCS and MCS values were observed in both groups. The scores for all domains reached the levels of the US normal population. Unfortunately, the authors did not analyze the dynamics of the SF-36 scores over time, so it is unclear at which stage after the procedure the improvement occurred [51]. Kolotkin et al. [52] reported the results of the Utah Study, a 12-year prospective cohort study to evaluate the trajectory and durability of HRQOL changes after bariatric treatment. The study included 418 gastric bypass patients and two non-surgical groups, persons who sought but did not have surgery (*n* = 417) and individuals with severe obesity who did not seek surgery (*n* = 321). HRQOL was measured at baseline and 2, 6 and 12 years post-surgery with two scales, IWQOL-Lite and SF-36. In the surgery group, both IWQOL-Lite scores and PCS values increased significantly from baseline to year 2 and then decreased slightly at 2–6 and 6–12 years, but still remained significantly higher than at the baseline and higher than in both non-surgical groups. In contrast, a slight improvement in MCS values observed at year 2 was not maintained at 6 and 12 years post-procedure. Changes in the BMI of the surgical group at 2–12 years post-procedure correlated inversely with IWQOL-Lite and PCS, but not with MCS changes [52]. Rolim et al. [53] used BAROS to evaluate the outcomes of RYGB in 42 patients ten years after the surgery. The majority of the patients (54.8%) qualified the outcome as “good”, with the proportions of “failures”, “fair”, and “very good” outcomes being 4.8%, 31.0% and 9.5%, respectively [53]. Galli et al. [54] evaluated long-term HRQOL ten years after biliointestinal bypass (BIB) surgery. The study included 90 patients interviewed by phone. HRQOL was determined with the SF-36. The results were compared with the reference values for the Italian healthy and general population. Compared with the general/healthy population, surgical patients were shown to present with significantly lower values of nearly all SF-36 domains. Up to 64.4% of the respondents declared that they were highly satisfied with the outcome of the BIB procedure. This subset of patients presented with higher scores for some of the SF-36 dimensions, namely physical functioning, vitality and general health perception [54].

Recently, Askari et al. [55] used BAROS to analyze HRQOL at least ten years after laparoscopic RYGB. The analysis included 92 out of 104 patients who underwent laparoscopic RYGB during the study period and completed a median follow-up of 130 months. The vast majority of the study patients had baseline BMI over 40 kg/m^2^. At the end of the follow-up period, participants reported generally feeling better, engaging in more physical activity, having more satisfactory social contacts, a better ability to work and a healthier approach to food. No significant change was, however, observed in sexual satisfaction scores. More than half of the patients (53.2%) reported a “good”, “very good”, or “excellent” outcome, whereas others assessed the result of RYGB as “fair” (26.1%) or “failure” (20.7%) [55]. 

### 3.2. Studies Analyzing the Surrogate Endpoints of HRQOL

Neovius et al. [56] analyzed two surrogate markers of HRQOL, suicide and non-fatal self-harm rates, in two historical cohorts of Swedish patients subjected to bariatric procedures. One of those cohorts, participants of the previously mentioned SOS study [43], included patients operated on in 1987–2001 and, hence, with at least nine years of follow-up available. The cohort consisted of 2010 patients subjected to bariatric procedures, vertical-banded gastroplasty (*n* = 1369), gastric banding (*n* = 376) or gastric bypass (*n* = 265), and 2037 patients who received a non-surgical intervention. During 68,528 person-years (median 18 years, interquartile range, IQR: 14–21), suicides or non-fatal self-harm events occurred significantly more often in the surgical than non-surgical group (87 vs. 49, adjusted hazard ratio, aHR = 1.78, 95% confidence interval, 95% CI: 1.23–2.57, *p* = 0.0021). When the results were stratified according to the bariatric procedure type, the highest risk of suicide/non-fatal self-harm was identified for gastric bypass (aHR = 3.48, 95% CI: 1.65–7.31, *p* = 0.0010), followed by gastric banding (aHR = 2.43, 95% CI: 1.23–4.82, *p* = 0.011) and vertical banded gastroplasty (aHR = 2.25, 95% CI: 1.37–3.71, *p* = 0.0015). Based on those findings, the authors concluded that bariatric treatment poses a risk of suicide or non-fatal self-harm, but the absolute value of that risk is low and does not justify a general discouragement of this kind of treatment [56].

The same group [57] analyzed associations of bariatric surgery with changes in interpersonal relationship status among the participants of the SOS study. Information about interpersonal relationship status was obtained with a questionnaire. The SOS cohort included in this analysis consisted of 1958 patients after bariatric surgery: gastric banding (*n* = 368), vertical banded gastroplasty (*n* = 1331) or gastric bypass (*n* = 259), along with 1912 matched obese controls receiving usual obesity care. The study participants were followed up for a median of 10 years (range 0.5–20 years). Bariatric surgery turned out to be associated with increased incidence of divorce/separation for those in a relationship (aHR = 1.28, 95% CI: 1.03–1.60, *p* = 0.03) and increased incidence of marriage or new relationship (aHR = 2.03, 95% CI: 1.52–2.71, *p* < 0.001) in those who were unmarried or single at the baseline [57]. Legenbauer et al. [58] analyzed another surrogate marker of HRQOL, body image dissatisfaction (BID) and body avoidance (BA) nine years after bariatric treatment using a cross-sectional dataset from the follow-up assessment of the Essen-Bochum Obesity Treatment Study (EBOTS). The analysis included 291 participants of the EBOTS, and among them, 78 were bariatric surgery patients, 124 were patients enrolled to a conventional treatment program and 83 were persons with obesity not seeking treatment. BID and BA at nine years were assessed via silhouette scales adapted for use in samples with obesity; additionally, BID was assessed retrospectively to obtain baseline values. An improvement in BID was observed in surgical patients but not in the other two groups. Current BID and BA correlated positively with current body weight, as well as with anxiety and depression. Regardless of treatment, an improvement in BID from the baseline was associated with successful weight loss. The results of that study highlight the role of various body image components as determinants of mental health, and thus, probably also HRQOL [58]. Mabey et al. [59] analyzed the mediators of suicidality in 131 patients 12 years after bariatric surgery (RYGB) and 205 individuals with severe obesity who did not undergo surgical treatment. Suicidality was assessed with the Suicide Behaviors Questionnaire—Revised, with metabolic health and HRQOL (SF-36 and IWQOL-Lite scores) at 0–2 and 2–6 years as exploratory variables. The study demonstrated that individuals undergoing bariatric treatment presented with higher suicidality at 12 years, which was mediated by lower improvements in MCS and PCS components of the SF-36. These findings constitute another argument for the monitoring of HRQOL in patients after bariatric treatment [59].

While bariatric surgery candidates reported impaired HRQOL presurgically, their HRQOL improved considerably after bariatric surgery. Our study shows that in 10 follow-ups, BMI was reduced, while weight loss was correlated with quality of life. The improvement in QOL scores correlated closely with the satisfaction with weight loss.

## 4. Discussion

This review demonstrated that HRQOL after bariatric surgeries generally tends to improve over the first 1–2 years after the procedure and then deteriorates again. Based on the results of a few studies with several follow-up points, the decrease occurs within initial 5–6 years after bariatric treatment, and the HRQOL scores at 9–12 years generally do not differ significantly from those observed at five years [43,47,48,50,52]. Despite the decrease in the HRQOL scores, in 8 out of 10 studies, their values at 9–12 years were still significantly higher than at the baseline [43,44,47,48,49,50,51,52], which implies that bariatric treatment had a persistent beneficial effect on the quality of life.

In a few studies [45,49,51,54], the HRQOL scores of patients after bariatric surgeries were compared with population norms; in some of those studies, HRQOL scores of bariatric patients at the end of the long-term follow-up were similar to those of the general population [45,51], but according to some authors [49,54], the patients’ scores were lower than the populational ones. These discrepancies are likely related to the baseline levels of HRQOL in patients qualified for bariatric treatment. Unfortunately, the authors of most of the studies included in this review did not provide information about the magnitude of HRQOL deterioration before treatment. However, based on other characteristics of included patients, their HRQOL scores were likely to vary from study to study. Some studies exclusively included patients with BMI 30–35 kg/m^2^ [45], whereas other studies analyzed subjects with BMI up to 60 kg/m^2^ [51]. Since pretreatment BMI is an established determinant of baseline HRQOL before bariatric treatment [22], some patients might present with a relatively worse baseline quality of life, and hence, their post-treatment HRQOL was less likely to normalize. This, in turn, implies that in future studies analyzing the trajectory of HRQOL after bariatric treatment, this parameter should be normalized for the baseline quality of life, as has been performed by Herpertz et al. [48].

Some studies included in this review [44,49,52] demonstrated that the degree of HRQOL improvement was proportional to the level of weight reduction or satisfaction with the weight reduction after bariatric treatment. This observation, consistent with the results of studies with shorter follow-up periods (as reviewed by [22]), might explain why, after the initial increase, the HRQOL of bariatric patients deteriorated again. Usually, during the first years after bariatric surgery, patients tend to be more compliant with the recommendations regarding diet and lifestyle. Then, their compliance naturally decreases with time, which is reflected by body weight gain and the resultant deterioration of HRQOL [24].

However, based on this review, an alternative explanation might also exist. In some studies using the SF-36 [47,48,52], the values of MCS were shown to deteriorate faster than those of PCS, and unlike the latter, returned to their baseline levels at the end of the follow-up period. This implies that while the physical wellbeing of patients after bariatric treatment may show persistent improvement, the same is not necessarily true for their mental wellbeing. Obviously, based on the pooled data, one cannot state whether this phenomenon occurs in all individuals. However, the results of some studies included in this review, using surrogate markers of HRQOL, imply that some fraction of patients after bariatric treatment might show some disturbances of mental health, e.g., increased suicidality, neuroticism, fear of intimacy, anxiety and depression, lowered sense of control and self-esteem [47,48,56,59]. It cannot be excluded that, due to those underlying problems, such patients are less compliant with postoperative recommendations and, within the mechanism of a vicious circle, regain weight faster, which leads to further deterioration of their HRQOL and worse outcomes of treatment. Identifying such patients before bariatric treatment or in the early postoperative period and providing them with a targeted psychological intervention would likely further improve the outcomes of the surgeries, not only in the context of HRQOL. However, this hypothetical mechanism should be verified empirically in a prospective study analyzing the psychological predictors of the bariatric treatment outcomes.

Regarding the use of obesity/procedure-specific HRQOL measures, we identified only a few studies using this kind of instrument, most often BAROS [44,46,53,55]. Unfortunately, BAROS scores are not a purely HRQOL measure but incorporate some physician-reported outcomes [3]. The authors of all but one study [55] reported only the overall BAROS score at the end of the follow-up period, rather than changes in specific HRQOL domains, which makes the results less applicable from the perspective of this review. Further, a few other studies (e.g., Martikainen et al. [61]) reported averaged BAROS scores for patients with variable duration of follow-up without distinguishing between the long- and short-term outcomes.

Several systematic reviews and meta-analyses have demonstrated that determination of HRQOL after bariatric treatment can be hampered by low-quality evidence, with only a few randomized clinical trials and well-designed prospective studies addressing this issue [20,22,25]. We observed a similar problem in the subset of studies included in this review. As shown in Table 1, only four studies [43,45,48,51] satisfied the criteria mentioned above. Furthermore, HRQOL was an additional outcome measure in most studies, and as a result, information about this parameter was not detailed enough. The increasingly recognized importance of HRQOL and other patient-reported outcomes [22] warrants further insight into those parameters optimally through large long-term randomized controlled trials and well-designed prospective studies. A limitation in our study at the audit and results level may be the risk of bias, and at the review level, it may be the incomplete search for identified studies or reporting bias. The results of this review indicate that bariatric treatment has a significant beneficial effect on the long-term assessment of quality of life.

## 5. Conclusions

In conclusion, bariatric treatment seems to provide a persistent benefit in terms of HRQOL, especially its physical component score. Due to psychological predispositions, some patients appear to be less likely to benefit from bariatric treatment, whether in terms of HRQOL or bodyweight reduction. The early identification of such patients and providing them with holistic care, including psychological intervention, would likely further improve the outcomes of bariatric treatment.

## Figures and Tables

**Figure 1 ijerph-19-09078-f001:**
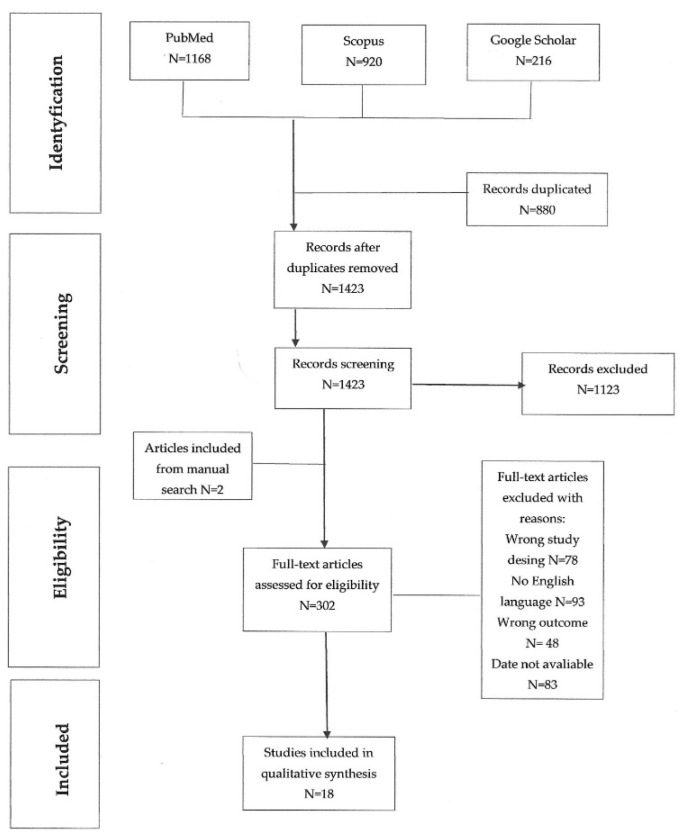
Flow chart of data extraction throughout the databases.

**Table 1 ijerph-19-09078-t001:** Methodology of the studies included in this review.

Authors	Type of the Study	Patients ^1^	BMI	HRQOL Determination Time	HRQOL Instruments
Karlsson et al. [43]	Non-randomized clinical trial	655/851 bariatric patients, 621/852 patients treated conventionally	Before: Men BMI ≥ 34, Women BMI ≥ 38 After bariatric intervention weight loss of approximately 10%	Baseline, 0.5, 1, 2, 3, 4, 6, 8 and 10 years	Current health scale, short version of the Mood Adjective Check List, Hospital Anxiety and Depression Scale (HADS), Social Interaction category from the Sickness Impact Profile, Obesity-Related Problems Scale
Kinzl et al. [44]	Mail survey	112/180 patients after laparoscopic gastric banding	EWL mean loss of overweight was 30.6%	Mean 10 years (range 9–12 years)	Bariatric Analysis and Reporting Outcome System (BAROS) and Bariatric Quality of Life Index
O’Brien et al. [45]	Randomized clinical trial	27/40 bariatric patients, 10/40 recipients of non-surgical intervention	EWL mean loss of overweight was 63.4%	Baseline, 2 and 10 years	36-Item Short Form (SF-36)
Aarts et al. [46]	Prospective cohort study	193/201 patients after laparoscopic adjustable gastric banding	Two-thirds of patients had an average weight loss of EWL > 50%	10 and 14 years (mean 13.6 years)	BAROS
Canetti et al. [47]	Non-randomized trial	36 bariatric patients, 34 participants of a weight loss program	After bariatric intervention weight loss of approximately 27%	Baseline, 1 and 10 years	SF-36, Mental Health Inventory (MHI), Neuroticism Scale of the NEO Five-Factor Inventory, Fear of Intimacy Scale, Shapiro Control Inventory
Herpertz et al. [48]	Prospective cohort study	84/152 bariatric patients, 126/249 participants of a weight loss program, 83/128 untreated obese controls	Before mean BMI was 50.7 kg/m^2^ (men and women) After bariatric intervention mean weight loss was 39.4 kg/m^2^(men and women)	Baseline, 1, 2, 4 and 9 years	SF-36, HADS, Rosenberg Self-Esteem Scale
Aasprang et al. [49]	Prospective cohort study	35/50 patients after biliopancreatic diversion with duodenal switch	Before mean BMI was 51.7 kg/m^2^ (men and women) After bariatric intervention mean weight loss was 32.9 kg/m^2^ (men and women)	Baseline, 1,2, 5 and 10 years	SF-36
Strain et al. [50]	Prospective cohort study	30/43 patients after biliopancreatic diversion with duodenal switch	Before mean BMI was 51.9 kg/m^2^ (men and women) After bariatric intervention mean weight loss was 32.0 kg/m^2^ (men and women)	Baseline, 1, 3, 5, 7 and 9 years	SF-36, Impact of Weight on Quality of Life—Lite (IWQOL-Lite), Beck Depression Inventory
Nguyen et al. [51]	Randomized clinical trial	46/111 patients after laparoscopic gastric bypass, 49/86 patients after laparoscopic gastric banding	Before mean BMI was 46.5 ± 5.6 kg/m^2^ After bariatric intervention mean weight loss was 37.5 ± 19.4 kg/m^2^ (men and women)	Baseline, 10 years	SF-36
Kolotkin et al. [52]	Prospective cohort study	260/418 gastric bypass patients, 242/417 controls who sought but did not have surgery, 235/321 controls with severe obesity who did not seek the surgery	BMI baseline 47.2 kg/m^2^	Baseline, 2, 6 and 12 years	SF-36, IWQOL-Lite
Rolim et al. [53]	Observational study	42 patients after Roux-en-Y gastric bypass	After bariatric intervention weight loss of approximately 22.3%	10 years	BAROS
Galli et al. [54]	Phone survey	90 patients after biliointestinal bypass surgery	Mean BMI 10-year follow-up 29.45 (HSG), 35.1 (LSG)	10 years	SF-36
Askari et al. [55]	Observational study	92/104 patients after laparoscopic Roux-en-Y gastric bypass	The pre-operative BMI dropped from a median of 46.9 to 37.3	Median 130 months	BAROS
Neovius et al. [56]	Retrospective analysis	2010 bariatric patients, 2037 recipients of a non-surgical intervention	Before mean BMI was 50.41 kg/m^2^ After bariatric intervention mean weight loss was 39.2 kg/m^2^ (men and women)	Median 18 years (interquartile range 14–21 years)	Suicide and non-fatal self-harm rates
Bruze et al. [57]	Retrospective analysis	1958 bariatric patients, 1.912 recipients of a non-surgical intervention	Before mean BMI was 42.2 kg/m^2^	Median 10 years (range 0.5–20 years)	Change in interpersonal relationship status
Legenbauer et al. [58]	Retrospective analysis	78 bariatric patients, 124 participants of a non-surgical program, 83 obese persons not seeking treatment	BMI Baseline 50.41 kg/m^2^, BMI follow-up 39.27 kg/m^2^	9 years	Body image dissatisfaction and body avoidance
Mabey et al. [59]	Prospective cohort study	131 patients after Roux-en-Y gastric bypass, 205 obese persons who did not undergo surgical treatment	12-year follow-up mean BMI 34.28 ± 8.06 kg/m^2^	baseline, 2, 6 and 12 years	SF-36 (as an explanatory variable), Suicide Behaviors Questionnaire-Revised
Felsenreich et al. [60]	Prospective study	65 patients after SG	BMI Baseline 48.7 ± 9.1 kg/m^2^, BMI follow-up 35.5 ± 6.7 kg/m^2^	Follow-up of ≥10 years	Bariatric Analysis and Reporting Outcome System (BAROS), Reflux Symptom Index (RSI), Gastrointestinal Quality of Life Index (GIQLI), Bariatric Quality of Life Index (BQL), and Short Form 36 (SF36)

^1^ Whenever the number of patients is expressed as a ratio, it should be interpreted as the number of patients available at the last follow-up/the number of patients available at the baseline (enrolled).

## Data Availability

Not applicable.

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
