# Peer review of "Quality of Life after Bariatric Surgery—A Systematic Review"

_ijerph, 2022, doi:10.3390/ijerph19159078_

Round 1

Reviewer 1 Report

The manuscript titled “Quality of life after bariatric surgery by Regina Sierżantowicz et al. is comprehensive and well written. However, it needs minor revision.

  1. In Table 1, the authors have described the methodology of the studies. It would be good if the authors also add detail about the average BMI of the patients during the assessment.
  2. In Table 1, Mabey et al. reference row font sizes are different from other content in the table.

Author Response

Thank you for the reviews and suggestions for improving our paper work

1. In Table 1, the authors have described the methodology of the studies. It would be good if the authors also add detail about the average BMI of the patients during the assessment.

Response: Corrected, we have added BMI details in the table 1.

2.In Table 1, Mabey et al. reference row font sizes are different from other content in the table.

Response: Corrected

English language was edited, certificate attached

Reviewer 2 Report

Thank you for the opportunity to review the manuscript, "Quality of life after bariatric surgery" submitted to the International Journal of Environmental Research and Public Health. Bariatric surgery, obesity reduction, and health related quality of life are important areas for good quality systematic reviews.

As there have been multiple reviews, including metasynthesis, conducted on this topic (see examples at close of summary), the importance of the current review needs to be stated. Some of the previously published reviews are cited in the manuscript, but not in the introduction for the context of prior evidence. For this reason, the rationale for conducting yet another systematic review needs to be clearly stated and justified.

The review provides this statement "The aim of this paper is to review sparse evidence regarding HRQOL in patients who underwent bariatric surgery at least nine years earlier." However, there is not a rationale with justification for selecting the nine year cut-point for the inclusion criteria.

There are multiple limitations that need to be addressed in the methods section from the search strategy to process for analysis based on the 2009 PRISMA guideline. As a note, the current guideline (see the 2020 revision) was not used, and the checklist was not provided with the current manuscript noting the page and paragraph (or line) numbers for each reporting criteria. This is very important as there appears to be some missing studies from the review. I am aware of at least one study from my previous work in this area (Felsenreich et al., 2019). Due to the lack of information about the research question (as recommended by the PRISMA), I am unable to further assess this point.

In the next summary area, the critical comments and constructive criticisms describe in detail the multiple limitations of the current review. Although many of the limitations can be addressed with revision, the missing literature and lack of an assessment for the risk of bias will most likely not be correctable with revision. 

SUMMARY FEEDBACK

What type of review does this manuscript report? The file name indicates a systematic review, but this is not stated in the manuscript.

What was the rationale for only searching PubMed, Scopus and Google Scholar? Why were other databases specific to this topic not included such as CINAHL and PsycINFO?

The PRISMA is not a method for conducting a review. Instead, the PRIMSA is a guideline for the reporting of a systematic review. What was the specific method was used for the review?

The 2009 PRISMA guideline applied to this review is outdated. The new version of the PRISMA published in 2020 should be applied to this review. In addition, the completed checklist, page and line or paragraph number indicated for each criteria, should be provided as a supplemental file with the manuscript. 

Multiple criteria from the PRIMSA, version 2009, were not reported in this manuscript. Although there are at least 12 criteria not addressed in the manuscript, five major deficiencies are noted.

1. The title does not identify the study as a systematic review (2009, #1, Identify the report as a systematic review, meta-analysis, or both).

2. The research question with rationale for the criteria guiding the search is not provided in the introduction (2009, #6, Specify study characteristics (e.g., PICOS, length of follow-up) and report characteristics (e.g., years considered, language, publication status) used as criteria for eligibility, giving rationale).

2. The search strategy for the databases are not provided (2009, #8, Present full electronic search strategy for at least one database, including any limits used, such that it could be repeated). Note, the 2020 guideline requires all the database searchers to be disclosed.

3. The risk of bias was not assessed (2009, #12, Describe methods used for assessing risk of bias of individual studies, and how this information is to be used in any data synthesis).

4. The limitations are not explicated as required (2009, #25, Discuss limitations at study and outcome level (e.g., risk of bias), and at review-level (e.g., incomplete retrieval of identified research, reporting bias).

The results section does not present a synthesis of the included studies. Instead, the results are listed with summaries provided for each study included in the review. The studies are independently reported by those analyzing HRQOL, and those analyzing the surrogate endpoints of HRQOL without synthesis of the studies. The lack of synthesis limits the value of the results section.

EXISTING REVIEWS

Health-related quality of life in bariatric and metabolic surgery (Coulman et al., 2020) https://doi.org/10.1007/s13679-020-00392-z (Reference 22)

Long-term health-related quality of life in bariatric surgery patients: A systematic review and meta-analysis (Driscoll et al., 2016) https://doi.org/10.1002/oby.21322 (Reference 29)

Health-related quality of life after bariatric surgery: A systematic review of prospective long-term studies (Andersen et al., 2015) https://doi.org/10.1016/j.soard.2014.10.027 (Reference 28)

Becoming physically active after bariatric surgery is associated with improved weight loss and health-related quality of life (Bond et al., 2009) https://doi.org/10.1038/oby.2008.501 (Not referenced).

STUDY NOT INCLUDED IN REVIEW

Felsenreich, D. M., Prager, G., Kefurt, R., Eilenberg, M., Jedamzik, J., Beckerhinn, P., Bichler, C., Sperker, C., Krebs, M., & Langer, F. B. (2019). Quality of life 10 years after sleeve gastrectomy: A multicenter study. Obesity Facts, 12(2), 157–166. https://doi.org/10.1159/000496296

Author Response

Thanks for the reviews and suggestions for improving our manuscript

  1. The title does not identify the study as a systematic review (2009, #1, Identify the report as a systematic review, meta-analysis, or both).

Response: We identify the study as a systematic review.

  1. The research question with rationale for the criteria guiding the search is not provided in the introduction (2009, #6, Specify study characteristics (e.g., PICOS, length of follow-up) and report characteristics (e.g., years considered, language, publication status) used as criteria for eligibility, giving rationale).

Response:

The search limits were defined as ‘English’ (language), and ‘the beginning of a given database through 31 October 2021’ (publication date) between 2007-2021.

The study used the PICO procedure, taking into account obesity with BMI> 30 kg / m2 and qualification for the bariatric procedure, performing a bariatric procedure, treatment in the control group of a non-bariatric procedure and the result, i.e. assessment of quality of life and weight loss.

  1. The search strategy for the databases are not provided (2009, #8, Present full electronic search strategy for at least one database, including any limits used, such that it could be repeated). Note, the 2020 guideline requires all the database searchers to be disclosed.

Response:

All randomized clinical trials, prospective, retrospective, cross-sectional and observation-al studies conducted in human patients receiving bariatric treatment (1), with HRQOL (2) and related parameters (body image, suicidality, social relationships, etc.) considered as out-come measures at least nine years after the procedure (3) and patients had obesity, (4) were included in the review.

The keywords used to search for the data were bariatric surgery, obesity, and quality of life, and were observed for ≥9 years.

  1. The risk of bias was not assessed (2009, #12, Describe methods used for assessing risk of bias of individual studies, and how this information is to be used in any data synthesis).

Response: Corrected and added:

Methodological quality assessment of included studies was performed by 2 authors (RS and JRŁ independently according to the Newcastle–Ottawa Scale (NOS) on diferent typ of studies (randomized and non-randomized clinical trial, prospective cohort study, observational study, mail and phone survey). Included studies were assessed items, including representativeness, patient selection, ascertainment, mention of conflict of interest, comparability, outcome assessment, follow-up length, and follow-up adequacy, with a maximum score of 9. Comparative studies with a score≥6 were considered of high quality (low risk of bias) while studies with a score<6 were considered of moderate or low quality (high risk of bias).

  1. The limitations are not explicated as required (2009, #25, Discuss limitations at study and outcome level (e.g., risk of bias), and at review-level (e.g., incomplete retrieval of identified research, reporting bias).

Response:

Added: A limitation in our study at the audit and results level may be the risk of bias, and at the review level, it may be incomplete search for identified studies or reporting bias.

The results section does not present a synthesis of the included studies. Instead, the results are listed with summaries provided for each study included in the review. The studies are independently reported by those analyzing HRQOL, and those analyzing the surrogate endpoints of HRQOL without synthesis of the studies. The lack of synthesis limits the value of the results section.

Response:

Corrected and added: Our study shows that in 10 follow-ups, BMI was reducated, while weight loss was correlated with quality of life. The improvement in QOL scores correlated closely with the satisfaction with weight loss.

STUDY INCLUDED IN REVIEW

Felsenreich, D. M., Prager, G., Kefurt, R., Eilenberg, M., Jedamzik, J., Beckerhinn, P., Bichler, C., Sperker, C., Krebs, M., & Langer, F. B. (2019). Quality of life 10 years after sleeve gastrectomy: A multicenter study. Obesity Facts, 12(2), 157–166. https://doi.org/10.1159/000496296

Corrected and added:

English language was edited, certificate attached

Reviewer 3 Report

The systematic review “Quality of life after bariatric surgery “ here presented is well and fluently written. It analyzes effects along the time often overlooked. Great job

I have only two tiny observations:

Could you give some details about the exclusion criteria for the 1123 excluded please?

cit.
line 116-118: A total of 17 published studies were identified, among them ten analyzing HRQOL as an outcome measure (with additional surrogate measures or without) and four solely with the outcome measures considered surrogate endpoints of HRQOL (Table 1).

10+4 is not equal to 17, the 3 missing are not described in this sentence, please add few words also for those.

Author Response

Thank you for the reviews and suggestions for improving the manuscript

Could you give some details about the exclusion criteria for the 1123 excluded please?

Response:

Inconsistent reporting of weight loss or post-bariatric testing results, but not related to quality of life assessment.

cit.
line 116-118: A total of 17 published studies were identified, among them ten analyzing HRQOL as an outcome measure (with additional surrogate measures or without) and four solely with the outcome measures considered surrogate endpoints of HRQOL (Table 1).

10+4 is not equal to 17, the 3 missing are not described in this sentence, please add few words also for those.

Response:

Corrected and added: A total of 18 published studies were identified, among them fourteen analyzing HRQOL as an outcome measure (with additional surrogate measures or without) and four solely with the outcome measures considered surrogate endpoints of HRQOL. We changed the total number of studies at the suggestion of another reviewer

English language was edited, certificate attached

Round 2

Reviewer 2 Report

Thank you for the revised version of the manuscript. The current systematic review was conducted in October 2021, almost a year after the publication of the updated version of the PRISMA in 2020. Yet, the manuscript continues to present the outdated PRISMA statement published in 2009. As noted in the prior review, this is serious problem that needs to be addressed prior to publication of a systematic review.  Finally, the authors need to send the completed PRISMA checklist with the page and line number noted for each criteria. This is not an optional process for a systematic review stating the PRISMA was used to guide the review. Thanks in advance for your help with addressing this remaining limitation.

Author Response

Thank you very much for your substantive comments that will improve our manuscript. PRISMA 2020 Check-lists have been completed. 42 literature items were changed to:

Page, MJ.; Moher, D.; Bossuyt, PM.; Boutron, I.; Hoffmann, TC.; Mulrow, CD.; Shamseer, L.; Tetzlaff, JM.; Akl, EA.; Brenan, SE.; Chou, R.; Glanville, J.; Grimshaw, JM.; Hróbjartsson, A.; Lalu, MM.; Li, T.; Loder, ET.; Mayo-Wilson, E.; McDonald, S.; McGuiness, LA.; Lesley A Stewart, LS.; Thomas, J.; Tricco, AC.; Welch, VA.; Whiting, P.; McKenzie JE. The PRISMA 2020 statement: an updated guideline for reporting systematic reviews. 2021; 372 :n71 doi:10.1136/bmj.n71

Some information has been added to the text: line 12, 14, 15, 19, 20 to the abstract and line 107, 108, 413, 414, 415 to the main text.
